# Low credibility URL sharing on Twitter during reporting linking rare blood clots with the Oxford/AstraZeneca COVID-19 vaccine

**Ali Hobbs** *, Aisha Aldosery, Patty Kostkova

IRDR Centre for Digital Public Health in Emergencies (dPHE), University College London, London, United Kingdom

* ali.hobbs.20@alumni.ucl.ac.uk

**Data Availability Statement:** All relevant data are within the paper and its Supporting information files. The datasets have been anonymised.

## Abstract

The COVID-19 pandemic was accompanied by an "infodemic" of misinformation. Misleading narratives around the virus, its origin, and treatments have had serious implications for public health. In March 2021, concerns were raised about links between the Oxford/AstraZeneca (AZ) COVID-19 vaccine and recipients developing blood clots. This paper aims to identify whether this prompted any reaction in the diffusion of low-credibility COVID-19-relate information on Twitter. Twitter's application programming interface was used to collect data containing COVID-19-related keywords between 4th and 25th March 2021, a period centred on the peak of new coverage linking rare blood clots with the AZ vaccine. We analysed and visualised the data using temporal analysis and social network analysis tools. We subsequently analysed the data to determine the most influential users and domains in the propagation of low-credibility information about COVID-19 and the AZ vaccine. This research presents evidence that the peak of news coverage linking rare blood clots with the AZ vaccine correlated with an increased volume and proportion of low-credibility AZ-related content propagated on Twitter. However, no equivalent changes to the volume, propagation, or network structure for the full dataset of COVID-19-related information or misinformation were observed. The research identified RT.com as the most prolific creator of low-credibility COVID-19-related content. It also highlighted the crucial role of self-promotion in the successful propagation of low-credibility content on Twitter. The findings suggest that the simple approach adopted within the research to identify the most popular and influential sources of low-credibility content presents a valuable opportunity for public health authorities and social media platforms to develop bespoke strategies to counter the propagation of misinformation in the aftermath of a breaking news event.

## Introduction

The COVID-19 pandemic, a global outbreak of the SARS-CoV-2 virus, occurred between 11 March 2020 and 5 May 2023 [1]. COVID-19 has occurred in an age where social media platforms enable the mass dissemination and consumption of information. The platforms which enable the public to stay informed and connected have also amplified an "infodemic",

**Funding:** This research was partly funded by the UKRI NERC under the grant NE/T013664/1.

**Competing interests:** The authors have declared that no competing interests exist.

characterised by the spreading of low-credibility information, which risks undermining public health activities [2]. The volume of misinformation circulating about COVID-19 is a major threat to public health, negatively impacting on people's compliance with public health guidance, and their willingness to get vaccinated [3].

In March 2021, following reports of patients developing rare blood clots after receiving the Oxford/AstraZeneca (AZ) vaccine, numerous countries suspended their AZ vaccination programmes [4]. This attracted vast amounts of media attention, with mainstream media coverage peaking on 15th March 2021 [5]. Research indicates that the AZ controversy, and the subsequent suspension of AZ vaccination programmes in several countries, made modest, though significant, contributions to increased vaccine hesitancy in Europe in March 2021 [6].

Understanding information diffusion networks and identifying influential individuals spreading low-credibility information about COVID-19 is essential to enable the delivery of targeted public health interventions [7]. This research will investigate how the propagation of low-credibility COVID-19-related information was impacted by media coverage of potential links between the AZ vaccine and blood clots, and who the most influential figures were in the sharing of low-credibility information.

Whilst there is an abundance of literature examining the propagation and impact of low-credibility information on social media platforms [8, 9], there is a lack of research regarding the impact of major news events on the dissemination of COVID-19-related information and misinformation. There is also limited investigation into the most influential users and domains in the propagation of low-credibility COVID-19-related information. Understanding how misinformation sharing pathways are impacted by breaking news, and which users and domains are the most prolific sharers of low-credibility content, will helpfully inform actionable strategies for minimising the spread and impact of COVID-19-related misinformation.

We addressed these gaps by investigating changes in the diffusion of COVID-19 information and misinformation from 4th to 25th March 2021, when media coverage linking the AZ vaccine to blood clots peaked. Our analysis investigates changes in the volume, time to first retweet, and network properties of COVID-19- and AZ-related tweets as media coverage of the AZ vaccine peaks and subsides. It further identifies and investigates the most influential users and domains in the propagation of low-credibility information about COVID-19 and the AZ vaccine during this period.

## Methodology

### Data collection and pre-processing

From April 2020, Twitter's application programming interface (API) was used to enable automated non-stop retrieval of 1% of tweets containing keywords related to the COVID-19 pandemic (available in S1 Table) (Data collection ceased in February 2023 when Twitter, now X, ended free access to its API.). Data collection ceased in February 2023 when Twitter ended free access to its API [10]. The collection and subsequent analysis method used complied with the terms and conditions for the source of the data. Twitter handles were available within the dataset which could be used to identify individual Twitter accounts. Tweets containing URLs posted in the week before, during, and after media coverage linking the AZ vaccine to blood clots peaked in the UK (11th– 17th March 2021 [5]) were selected for analysis. Details of the study period are provided in Table 1.

To identify low-credibility information propagating on Twitter, this research focuses on the sharing of URL domains which have been identified as routinely publishing low-credibility information. This methodology has been widely adopted across the literature [11–17].

**Table 1. Details of the study period.**

| Period Name | Dates |
| --- | --- |
| Pre-Peak | 4th– 10th March 2021 |
| Peak | 11th– 17th March 2021 |
| Post-Peak | 18th– 25th March 2021 |

The Iffy+ Misinfo/Disinfo dataset (the Iffy+ List) is a list of domains that regularly publish misinformation or disinformation collated using lists of low-credibility sources from independent fact-checking organisations Media Bias/Fact Check (MBFC), FactCheck.org, and Politi-Fact; digital media organisation BuzzFeed; and online encyclopaedia Wikipedia [18]. The dataset verifies sources in 19 languages, including the seven featured in our keyword list (see S1 Table for those seven languages). This dataset was used to determine which of the URL domains can be classified as low-credibility. The Iffy+ List from March 2021 contains 812 domains. By extracting the tweets which direct to low-credibility domains, a dataset of tweets referencing low-credibility information (the Low-Credibility dataset) and a dataset of tweets containing URLs which were not judged to be low-credibility (the Residual dataset) were created. To enable interrogation of tweets which reference the AZ vaccine, a filter was applied to the COVID Corpus to identify those tweets containing keywords related to the AZ vaccine to produce a corpus of AZ-related tweets (keywords available in S1 Appendix). The data filtering is shown in Figs 1 and 2.

## Temporal analysis

To identify changes in the volume of relevant tweets posted across the study period, we produced timeseries of tweets posted daily. The data was further interrogated to create timeseries of potential exposure to relevant tweets, derived using the estimated audience size for each tweet posted [19]. This was done by summing the follower count of each tweet's author. This measure of potential exposure is neither an upper nor lower bound of the actual exposure, as exposure is possible outside a following relationship, and there is no guarantee that all followers read a user's tweets.

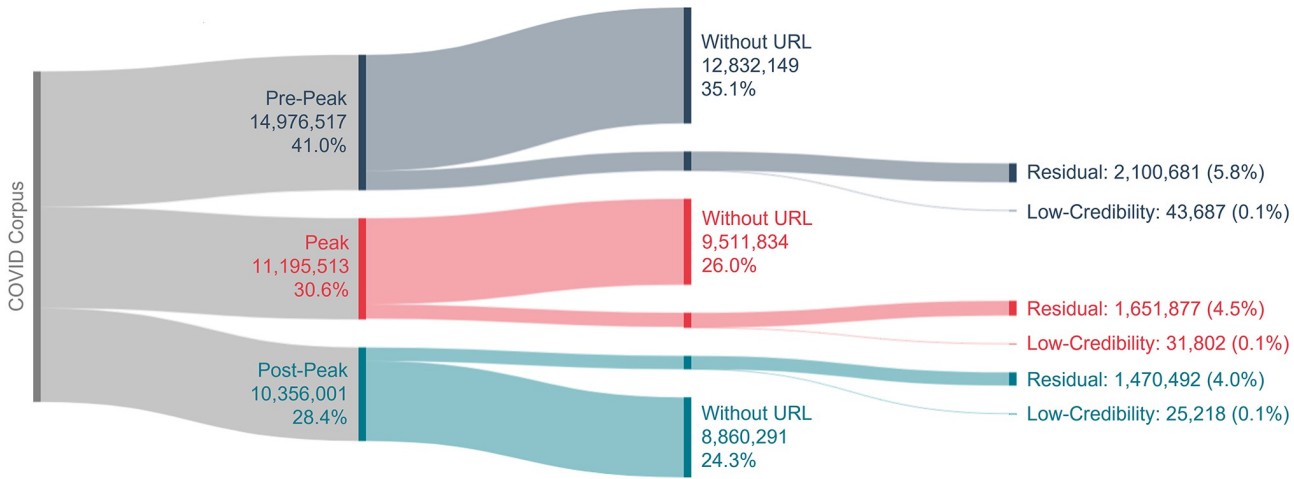

**Fig 1. Sankey diagram showing the filtering of the COVID datasets used for analysis.** Pre-Peak datasets are in navy, Peak datasets in red, and Post-Peak datasets in teal.

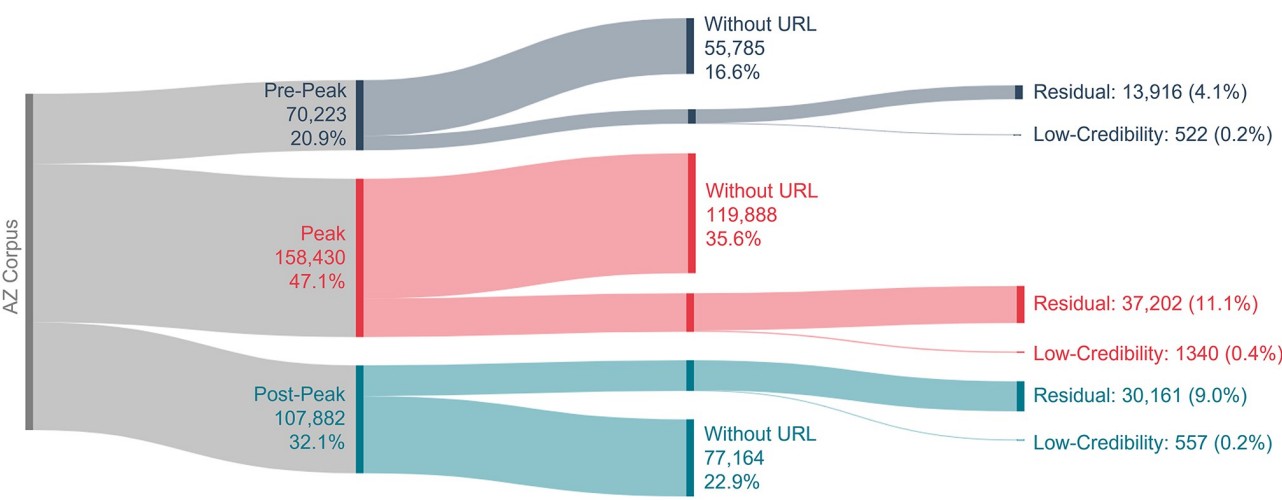

**Fig 2. Sankey diagram showing the filtering of the AZ datasets used for analysis.** Pre-Peak datasets are in navy, Peak datasets in red, and Post-Peak datasets in teal.

The speed of propagation of information was determined by identifying the time between a tweet being posted and receiving its first retweet. These time periods were plotted on an empirical cumulative distribution plot.

## Network analysis

To identify changes in the network properties of COVID-19-related information diffusion on Twitter during the study period, directed networks were created from each of the datasets using Python library NetworkX.

$$NetworkGraph = (Uw, E)$$

where $U$ is a set of those Twitter users who posted tweets in the study period,
$w$ is the weight of the node representing the sum of the number of retweets that user received for each tweet posted within the study period, and
$E$ is a set of directional edges representing a retweet, going from the originator of the post to the user who retweeted it.

In-built algorithms within NetworkX were used to determine the network properties required for analysis. Due to the size of the datasets being analysed, the choice of algorithms was made by balancing computational efficiency and accuracy.

Some network properties were not possible to investigate due to limitations of the API. Twitter's model of information sharing does not preserve the chain-sharing of information, omitting any intermediaries who facilitate the information sharing between the author and a user retweeting the post. This limitation was recognised by Kostkova, Mano, Larson & Schulz [7] and is visually represented in Fig 3.

This limitation removes some details about the mechanism by which information is shared. Instead, it creates two distinct types of nodes within a graph: "Authors" who are the originators of content (user A in Fig 3); and "Sharers" who share Authors' content with their followers via retweets (users B and C). This means that the networks produced for this research will have distinct hub-and-spoke structures, with Authors as hubs and Sharers as spokes.

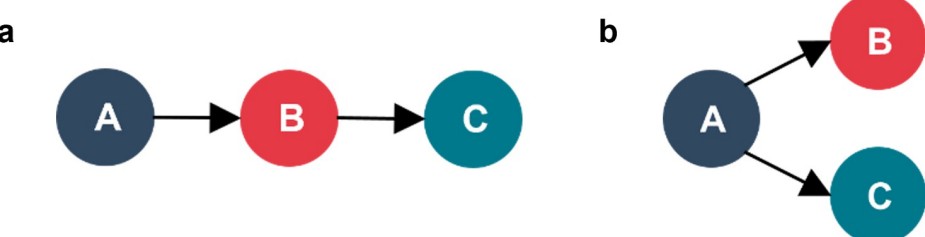

**Fig 3. Two networks representing information diffusion; 3a showing the true information sharing chain from the original author, user A, retweeted by user B, and subsequently retweeted by user C; and 3b showing Twitter's model of information sharing where user B's role as an intermediary is omitted.**

Despite this limitation, it is still possible to use networks to identify the most popular Authors, and where Authors are connected to the same Sharers it suggests commonality in readership, providing insights into sub-networks.

**Network composition.**   Network composition considers the size and make-up of the network. This was determined by identifying the number of nodes in the graph, as well as the number of Authors, Sharers, Author/Sharers, and Isolates. These classifications were determined by considering the degree of each node. In-degree is the number of inward edges a node has, and therefore represents the number of different Authors a Sharer has retweeted. Out-degree is the number of outward edges a node has, and therefore represents the number of different Sharers who have retweeted an Author's content.

Authors were defined as those users who have solely posted tweets. Sharers were defined as those users who have solely posted retweets. Author/Sharers are those users who are both originators of content and propagate other Authors' content. Isolates are nodes which are not connected to any other node within the graph.

$$Classification = \begin{cases} Author, if\ k_{out} > 0\ \text{AND}\ k_{in} = 0 \\ Sharer, if\ k_{in} > 0\ \text{AND}\ k_{out} = 0 \\ Author/Sharer, if\ k_{in} > 0\ \text{AND}\ k_{out} > 0 \\ Isolate, if\ k_{in} = k_{out} = 0 \end{cases}$$

where $k_{in}$ is the in-degree of the node being classified
$k_{out}$ is the out-degree of the node being classified

Networks were visualised using the network visualization software Gephi, using the ForceAtlas2 algorithm.

**Network structure.**   Measures of centrality were used to analyse the network structures associated with each network, as well as to identify the most influential nodes within each network.

In-degree and out-degree centrality were calculated for each of the directed networks using in-built algorithms within NetworkX. Out-degree centrality is the most direct measure of an author's popularity, showing the number of Sharers who have retweeted their content. In-degree centrality measures the variety in the Authors retweeted by a given Sharer.

Eigenvector centrality was also calculated for each network using NetworkX using undirected graphs. Eigenvector centrality considers each node's direct and indirect neighbours [20]. For directed graphs, due to Twitter's model of information sharing, there is limited

difference between measures of degree centrality and eigenvector centrality as Authors will only have indirect neighbours where their content was shared by another author. This is not a meaningful indicator of influence within the network. However, for undirected graphs, eigenvector centrality becomes a measure of audience cross-over between Authors. Undirected graphs with higher eigenvector centrality will be more interconnected than those with lower eigenvector centrality.

The distributions of each centrality measure were plotted, and the median and maximum values extracted for analysis.

## Domain and user analysis

Domain and user analysis were conducted to identify the most popular sources of low-credibility COVID-19- and AZ-related information on Twitter during the study period, and the most influential Twitter users sharing this content. This was the only part of the research where Twitter handles were interrogated.

The most popular domains were extracted from the Low-Credibility datasets to understand which are most frequently shared. For the COVID Corpus, those domains which were posted at least 1000 times in one of the week-long periods were plotted on a stacked bar chart. For the AZ Corpus, those domains which were posted at least 20 times were plotted.

Out-degree centrality was selected as the most appropriate measure of influence. The out-degree centrality from the Pre-Peak, Peak, and Post-Peak periods were summed to identify the top 100 Authors of content from each dataset. The account properties of these influential authors were then compared to the baseline properties of the active Twitter userbase posting about COVID-19 during the study period. The properties investigated were whether accounts were verified (showing they were authenticated by Twitter as an account of public interest), when the accounts were created, and the accounts' follower counts.

## Results

A total of 36.5 million COVID-19-related tweets were collected between 4 and 25 March 2021; 5.3 million of which (14.5%) contained URLs. 336,483 tweets referenced AZ (0.9% of all COVID-19-related tweets); 83,646 of which (24.9%) contained URLs. We identified 100,708 COVID-19-related tweets which referenced low-credibility domains (0.28% of all COVID-19-related tweets, and 1.9% of those containing URLs); and 2,419 AZ-related tweets which referenced low-credibility domains (0.72% of all AZ-related tweets, and 2.9% of those containing URLs). Fig 4 plots how the frequency of tweets varies across the study period. A scaling factor of 20 has been applied to the low-credibility trace for visibility.

Peaks in the volume of AZ-related tweets, and low-credibility domains, is visible during the peak of AZ media coverage. There is no notable change in the COVID-19 trace.

We produced word clouds of the most used words in COVID-19-related tweets on the 9 and 10 March 2021 where there is a prominent peak (Fig 5). Stop words have been filtered out.

"COVID relief", mentioned on both days, is a reference to the American Rescue Plan Act of 2021, a 1.9 trillion USD economic stimulus bill which passed through the House of Representatives on 10 March 2021, and was signed into law on 11 March 2021, the first anniversary of COVID-19 being declared a pandemic by the WHO [1, 21]. "Ericas Day" and "Elite ICU" are phrases used in a trending Twitter discussion about Nigerian actress Erica Ngozi Nlewedim [22]. These have erroneously been included in the dataset as "ICU" is one of our COVID-19 keywords. Further analysis of the tweets on these days is required to ascertain the cause of these prominent peaks.

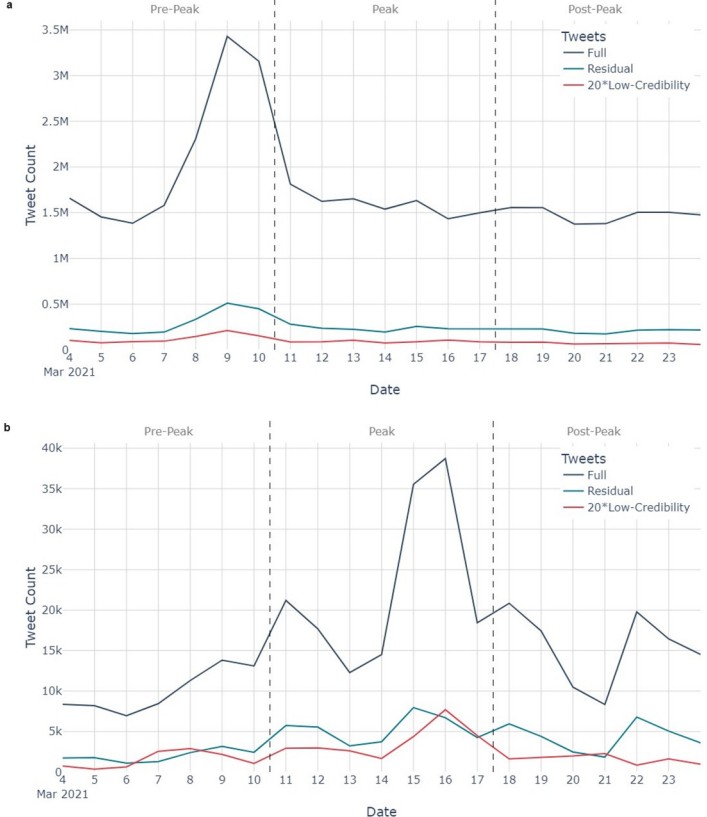

**Fig 4. A timeseries of tweets (navy), Low-Credibility tweets x20 (red), and Residual tweets (teal) over the study period for 4a the COVID-19-related tweets, and 4b the AZ-related-tweets.**

## Potential exposure

Fig 6 plots how the exposure varies across the study period. A scaling factor of 20 has been applied to the low-credibility trace for visibility. Table 2 shows the average potential readership for each of the datasets over the study period.

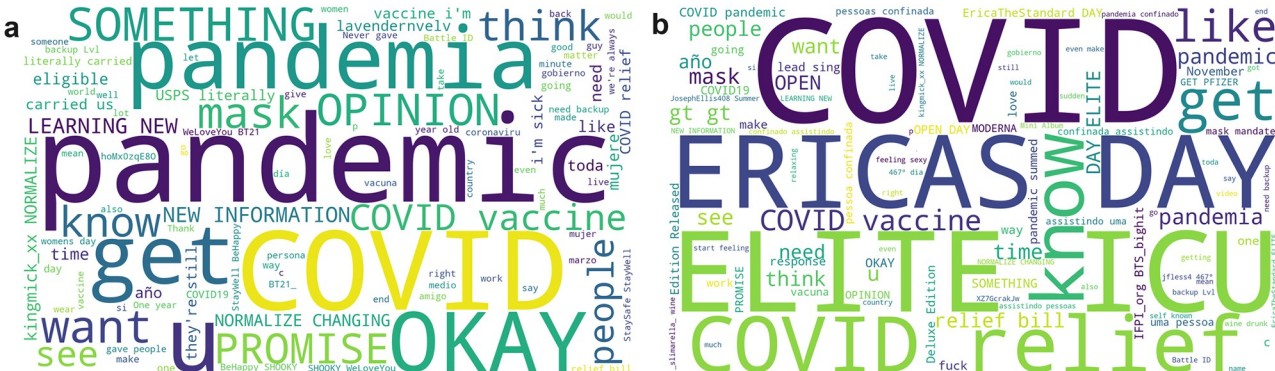

**Fig 5. Word clouds showing the most used words in tweets and retweets made on 5a 9 March 2021, and 5b 10 March 2021.**

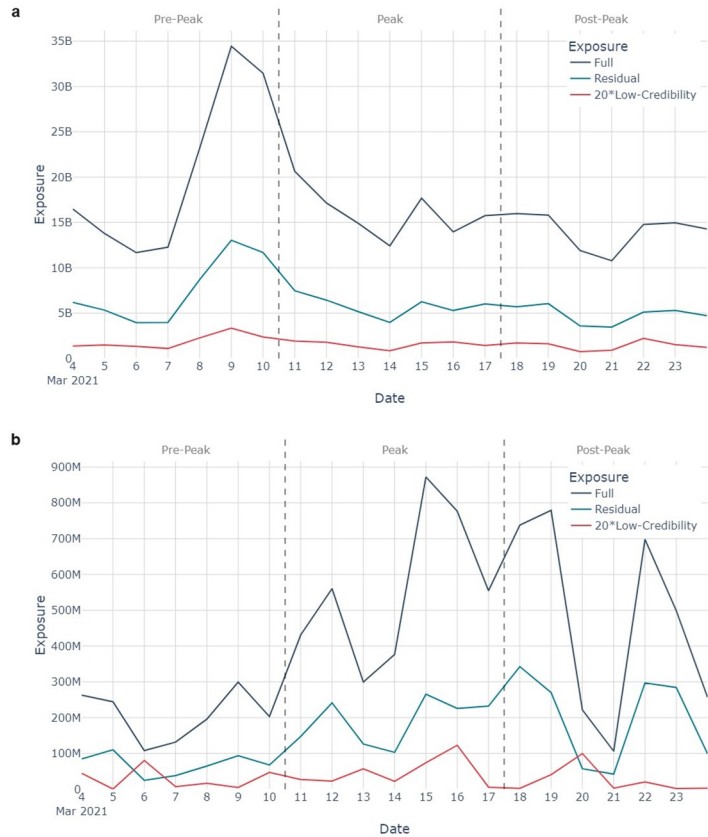

**Fig 6. A timeseries of potential exposure to tweets (navy), Low-Credibility tweets x20 (red), and Residual tweets (teal) over the study period for 6a the COVID-19-related tweets, and 6b the AZ-related-tweets.**

Whilst the exposure trace for COVID-19-related tweet follows broadly the same shape as the timeseries, URL-containing tweets enjoyed a greater readership than those without URLs (9,700 users/tweet for the full dataset; 24,000 users/tweet for the Residual dataset and 17,000 users/tweet for the Low-Credibility dataset). The difference in readership is statistically significant (Mann-Whitney U test, p-value < 0.05 when comparing the full dataset to both the residual and low-credibility datasets).

The exposure trace for AZ-related tweets notably deviates from the shape of the timeseries, particularly during the peak and post-peak periods. For example, on 19 March 2021 the average potential readership was 45,000 users/tweet compared to 26,000 users/tweet on average across the study period. The difference between the readerships is statistically significant (Mann-Whitney U test, p-value < 0.05). This suggests that on 19 March 2021, Twitter users with higher numbers of followers were posting more AZ-related content than was typical across the study period.

**Table 2. Average potential readership (users/tweet) ± standard distribution for each of the datasets across the study period.**

|       | Full Dataset      | Residual Dataset  | Low-Credibility Dataset |
|-------|-------------------|-------------------|-------------------------|
| COVID | 9,700 ± 250,000   | 24,000 ± 490,000  | 17,000 ± 220,000        |
| AZ    | 26,000 ± 530,000  | 40,000 ± 760,000  | 15,000 ± 160,000        |

### Time to first retweet

The speed of propagation of information was determined by the time elapsed between a tweet and its first retweet. This was conducted as a measure of engagement with COVID-19- and AZ-related content. The cumulative distribution plots for time elapsed is shown in Fig 7.

For each dataset, the time elapsed between a tweet and its first retweet was highest in the Pre-Peak period. There is little variation in the time to first retweet between the Peak and Post-Peak periods, except for the Low-Credibility AZ-related tweets where the median time elapsed was 2.30 minutes during the Peak period, compared to 30.5 minutes and 7.25 minutes during the Pre-Peak and Post-Peak periods respectively. However, despite these apparently stark

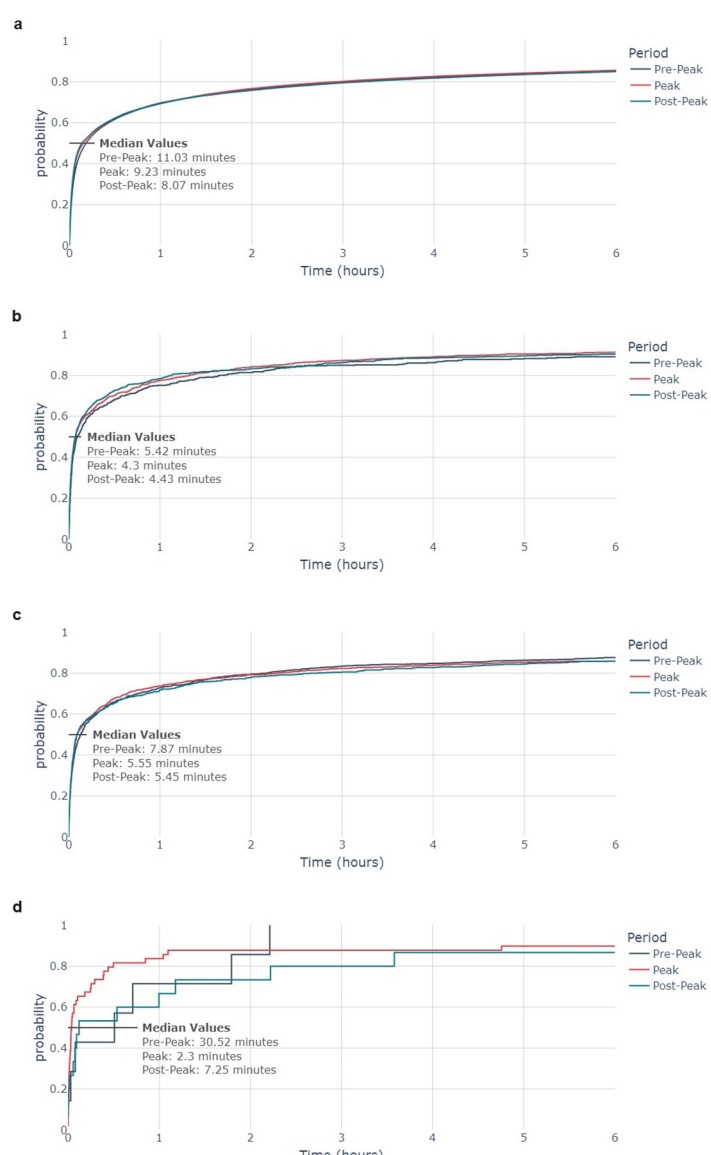

**Fig 7. Cumulative distributions of the time elapsed between a tweet being posted and receiving its first retweet for the 7a COVID Residual dataset, 7b COVID Low-Credibility dataset, 7c AZ Residual dataset, and 7d AZ Low-Credibility dataset.** Time elapsed for tweets posted in the Pre-Peak period are in navy, those for the Peak period are in red, and those for the Post-Peak period are in teal.

**Table 3. The percentage of tweets which were retweeted within the study period.**

|  |  | Pre-Peak | Peak | Post-Peak |
|---|---|---|---|---|
| Residual | COVID | 17.2% | 14.6% | 13.7% |
|  | AZ | 12.7% | 14.9% | 14.9% |
| Low-Credibility | COVID | 15.9% | 13.8% | 13.1% |
|  | AZ | 8.05% | 17.4% | 11.0% |

differences, the Kruskal-Wallis H-test indicates that there is no significant difference between the distributions (p-value > 0.05).

Table 3 shows the percentage of tweets which were retweeted within the study period. The Low-Credibility AZ dataset is the only one which shows notable variation across the three weeks, with 17.4% of tweets being retweeted in the Peak period, compared with 8.05% and 11.0% in the Pre-Peak and Post-Peak periods respectively. Therefore, during the peak of news coverage linking the AZ vaccine to blood clots, AZ-related tweets were substantially more likely to be retweeted than in the week before or afterwards. This indicates that there was increased engagement with AZ-related tweets which coincided with the peak news coverage, but that this quickly subsided.

## Network analysis

Network analysis was conducted to investigate how COVID-19-related information diffusion networks changed as negative media coverage of the AZ vaccine peaked and subsided in March 2021.

Networks were constructed for the Low-Credibility and Residual datasets derived from the COVID and AZ Corpora across each of the time periods. Fig 8 shows the networks graphs for the Low-Credibility COVID datasets and the Residual AZ datasets for the three time periods. Authors (originators of content) are in red; Sharers (those who propagate Authors' content via retweets) are in navy; Author/Sharers (those who both create original content and retweet others' content) are in teal; Isolates (those who are not connected to any other node within the graph) are in yellow. Isolates may represent Authors whose content has not been retweeted by any other users, or Sharers who have retweeted content that was posted prior to the study period.

These graphs are reminiscent of Smith et al.'s [23] broadcast networks, with distinct hub-and-spoke structures, with Authors or Author/Sharers as hubs, and Sharers as spokes. There are limited connections between nodes outside of these structures.

**Network composition.**   Table 4 shows the composition of the 12 networks constructed for this research, six of which are displayed in Fig 8 for illustrative purposes.

In the COVID Low-Credibility networks there are slightly higher proportions of Sharers and slightly lower proportions of Authors in the Pre-Peak period than the Peak or Post-Peak periods (7.82% Authors and 80.9% Sharers in the Pre-Peak period, compared to 9.27% and 9.82% Authors and 77.5% Sharers in the Peak and Post-Peak periods). This is shown visually by comparing Fig 8a–8c. In Fig 8a, there are fewer but larger clusters of nodes than in Fig 8b and 8c. This is indicative of a greater number of Sharers retweeting a smaller number of Authors' posts. This finding is replicated in the AZ Low-Credibility networks, where there are progressively higher proportions of Authors and lower proportions of Sharers as you progress through the study period.

The number of nodes (N) in the AZ Corpus during the Peak period is considerably higher than in the Pre-Peak period, and slightly higher than in the Post-Peak period (13,377, 33,444,

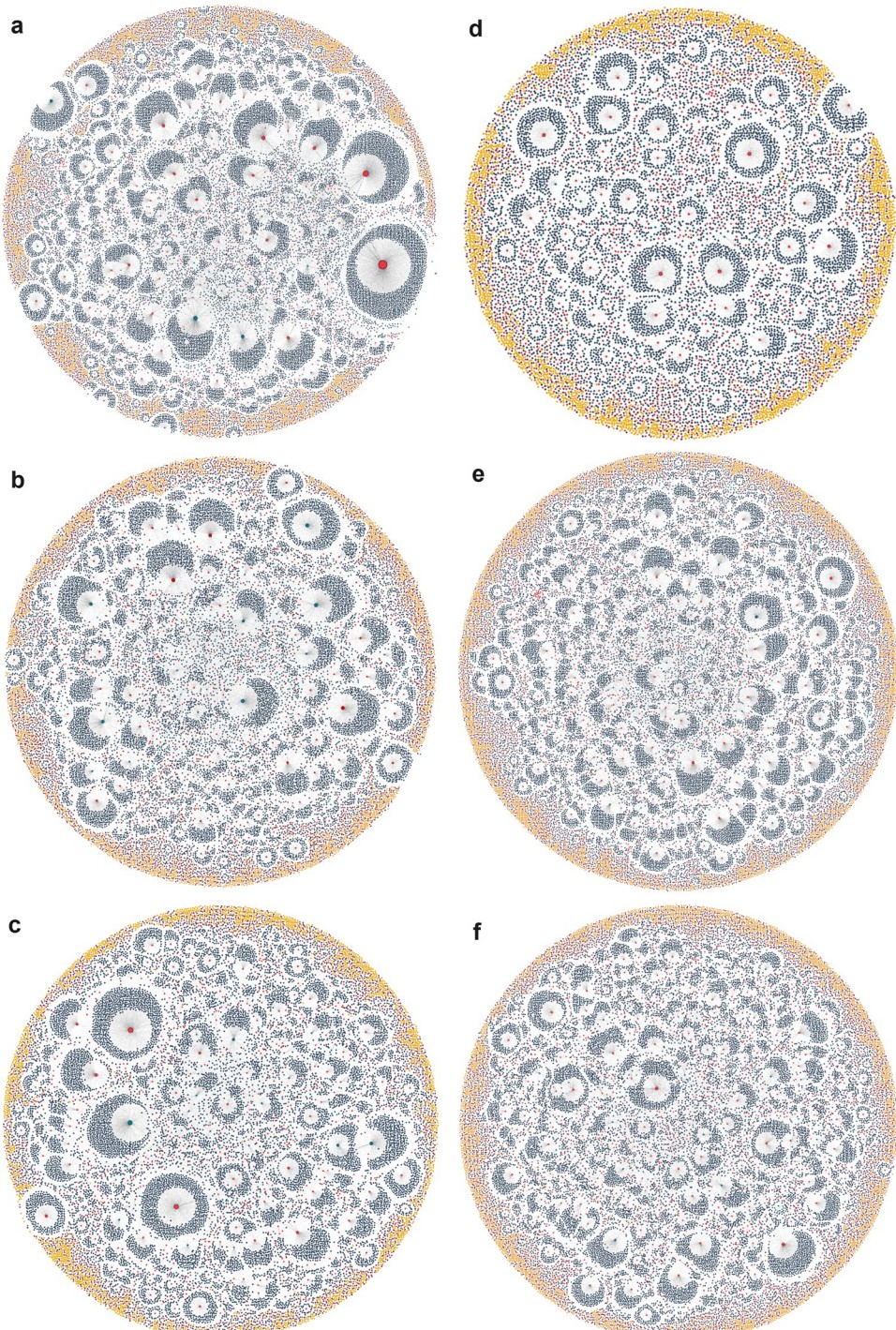

**Fig 8. Networks produced from the Low-Credibility COVID dataset in the 8a Pre-Peak, 8b Peak, and 8c Post-Peak period; and from the Residual AZ dataset from the 8d Pre-Peak, 8e Peak, and 8f Post-Peak period.** Authors are in red, Author/Sharers in teal, Sharers in navy, and Isolates in yellow.

**Table 4. Network composition properties for Residual and Low-Credibility datasets, showing the total number of nodes, Authors, Sharers, Author/Sharers, and Isolates.** With the exception of the number of nodes, each value is shown in absolute terms and as a percentage of the number of nodes in the graph.

| | | | N | $N_{Authors}$ (% of N) | $N_{Sharers}$ (% of N) | $N_{Author/Sharers}$ (% of N) | $N_{Isolates}$ (% of N) |
|---|---|---|---|---|---|---|---|
| COVID Corpus | Residual | Pre-Peak | 1,213,708 | 152,659 | 866,025 | 43,112 | 151,912 |
| | | | | (12.6%) | (71.4%) | (3.6%) | (12.5%) |
| | | Peak | 1,036,541 | 142,077 | 730,871 | 34,145 | 129,448 |
| | | | | (13.7%) | (70.5%) | (3.3%) | (12.5%) |
| | | Post-Peak | 933,590 | 130,136 | 653,092 | 29,111 | 121,251 |
| | | | | (13.9%) | (70.0%) | (3.1%) | (13.0%) |
| | Low-Credibility | Pre-Peak | 32,713 | 2,557 | 26,448 | 329 | 3,379 |
| | | | | (7.8%) | (80.9%) | (1.0%) | (10.3%) |
| | | Peak | 25,189 | 2,336 | 19,526 | 326 | 3,001 |
| | | | | (9.3%) | (77.5%) | (1.3%) | (11.9%) |
| | | Post-Peak | 21,004 | 2,063 | 16,282 | 192 | 2,467 |
| | | | | (9.8%) | (77.5%) | (0.9%) | (11.8%) |
| AZ Corpus | Residual | Pre-Peak | 13,337 | 1,656 | 9,152 | 63 | 2,466 |
| | | | | (12.4%) | (68.6%) | (0.5%) | (18.5%) |
| | | Peak | 33,444 | 3,708 | 25,210 | 315 | 4211 |
| | | | | (11.1%) | (75.4%) | (0.9%) | (12.6%) |
| | | Post-Peak | 27,833 | 3,226 | 20,687 | 260 | 3660 |
| | | | | (11.6%) | (74.3%) | (0.9%) | (13.2%) |
| | Low-Credibility | Pre-Peak | 530 | 43 | 416 | 1 | 70 |
| | | | | (8.1%) | (78.5%) | (0.2%) | (13.2%) |
| | | Peak | 1,330 | 138 | 1002 | 2 | 188 |
| | | | | (10.4%) | (75.3%) | (0.2%) | (14.1%) |
| | | Post-Peak | 573 | 76 | 401 | 0 | 96 |
| | | | | (13.3%) | (70.0%) | (0.0%) | (16.8%) |

and 27,833 for the Pre-Peak, Peak, and Post-Peak periods respectively). This is shown visually by comparing Fig 8d to 8e, where the density of nodes is notably greater. This reflects the findings from the temporal analysis regarding the higher proportion of tweets in the AZ Corpus occurring during the Peak period. However, as nodes reflect individual users, rather than tweets, this indicates that substantially more Twitter users were posting AZ-related tweets during the Peak period, rather than a similar userbase posting more tweets each.

**Network structure.** *Centrality.* Three different measures of centrality were applied to each network:

1. In-degree centrality ($c_{in}^{DEG}$), capturing the variety of Authors retweeted;

2. Out-degree centrality ($c_{out}^{DEG}$), capturing the popularity of Authors in the network;

3. Eigenvector centrality ($c^{EIG}$), capturing the interconnectivity of the network.

Median and maximum values for these centrality measures for each of the networks are in Table 5. Violin plots of the distribution of centrality measures for the Low-Credibility AZ-related tweets are shown in Fig 9.

For most of the networks, there are limited differences in centrality measures between time periods. However, for the AZ Low-Credibility dataset, there is a statistically significant difference in the distribution of in-degree centrality between the Peak and both the Pre-Peak and

**Table 5. Median and maximum centrality measures for the Residual and Low-Credibility datasets.**

| | | | $c_{in}^{DEG}$ | $c_{out}^{DEG}$ | $c^{EIG}$ |
|---|---|---|---|---|---|
| COVID Corpus | Residual | Pre-Peak | $8.2 \times 10^{-7}$ | 0.0 | $2.2 \times 10^{-11}$ |
| | | | 0.00098 | 0.0062 | 0.65 |
| | | Peak | $9.6 \times 10^{-7}$ | 0.0 | $2.1 \times 10^{-9}$ |
| | | | 0.00073 | 0.0060 | 0.65 |
| | | Post-Peak | $1.1 \times 10^{-6}$ | 0.0 | $6.9 \times 10^{-10}$ |
| | | | 0.00055 | 0.0067 | 0.53 |
| | Low-Credibility | Pre-Peak | $3.1 \times 10^{-5}$ | 0.0 | $1.8 \times 10^{-6}$ |
| | | | 0.00055 | 0.073 | 0.70 |
| | | Peak | $4.0 \times 10^{-5}$ | 0.0 | $1.5 \times 10^{-5}$ |
| | | | 0.00044 | 0.024 | 0.49 |
| | | Post-Peak | $4.8 \times 10^{-5}$ | 0.0 | $1.81 \times 10^{-8}$ |
| | | | 0.00052 | 0.042 | 0.71 |
| AZ Corpus | Residual | Pre-Peak | $7.5 \times 10^{-5}$ | 0.0 | $2.9 \times 10^{-45}$ |
| | | | 0.0011 | 0.021 | 0.65 |
| | | Peak | $3.0 \times 10^{-5}$ | 0.0 | $4.2 \times 10^{-8}$ |
| | | | 0.0010 | 0.013 | 0.50 |
| | | Post-Peak | $3.6 \times 10^{-5}$ | 0.0 | $1.2 \times 10^{-7}$ |
| | | | 0.0010 | 0.018 | 0.70 |
| | Low-Credibility | Pre-Peak | 0.0019 | 0.0 | $5.4 \times 10^{-5}$ |
| | | | 0.0038 | 0.20 | 0.71 |
| | | Peak | 0.00075 | 0.0 | $1.5 \times 10^{-17}$ |
| | | | 0.0023 | 0.091 | 0.70 |
| | | Post-Peak | 0.0017 | 0.0 | $3.9 \times 10^{-21}$ |
| | | | 0.0035 | 0.015 | 0.71 |

Post-Peak periods (Mann-Whitney U test, p-value < 0.05). There is also a statistically significant difference in the distribution of eigenvector centrality between the Pre-Peak and Peak periods (p-value < 0.05).

In-degree centrality is a measure of the variety in the Authors retweeted by the Sharers, and eigenvector centrality is a measure of interconnectivity of the network. Both findings are indicative of Sharers in the Pre-Peak period retweeting a greater variety of AZ-related content than in the Peak and Post-Peak periods. The differences in out-degree centrality are not statistically significant (p-value > 0.05).

## Domain and user analysis

Domain and user analysis were conducted to understand the most influential sources and users in the propagation of low-credibility COVID-19- and AZ-related information.

**Popular sources.** Fig 10 shows the nine most popular sources of low-credibility information in the COVID Corpus based on how many times they were tweeted or retweeted during the study period. Russian government-funded television news network RT.com is the low-credibility source most frequently referenced in each of the three time periods.

Fig 11 shows the most popular low-credibility domains within the AZ Corpus. Again, RT.com is the low-credibility source most frequently referenced in tweets and retweets. There is a significant increase in the number of times RT.com is referenced in the Peak period (647) compared to the Pre-Peak and Post-Peak periods (178 and 238 respectively). A similar pattern

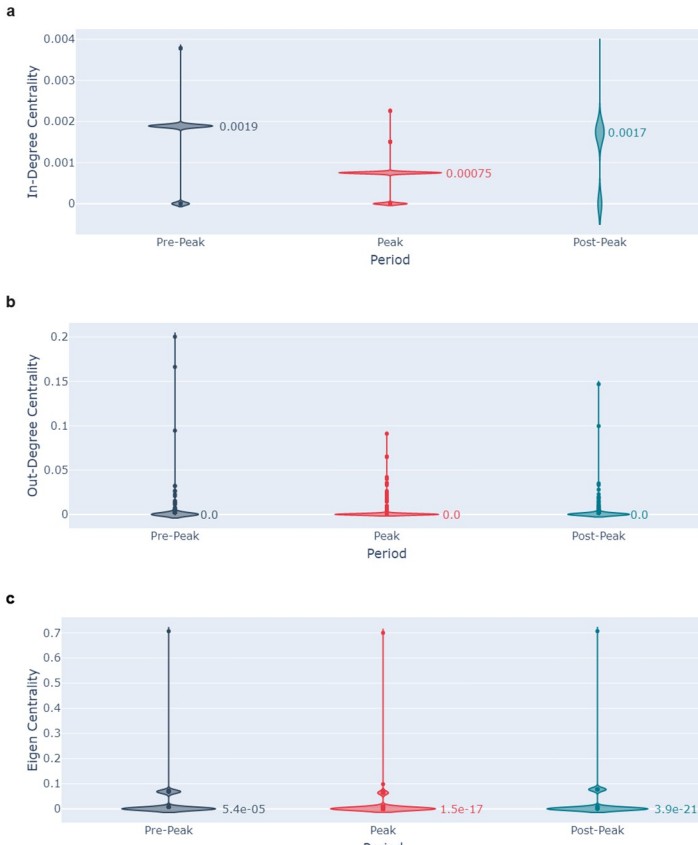

**Fig 9. Violin plots of the distribution of 9a the in-degree centrality, 9b the out-degree centrality, and 9c the eigenvector centrality for the AZ Low-Credibility dataset.** The distributions for the Pre-Peak period are in navy, those for the Peak period are in red, and those for the Post-Peak period are in teal.

is identified for childrenshealthdefence.org, an American activist group mainly known for anti-vaccine propaganda which has been identified as a key source of misinformation on vaccines [24]. Childrenshealthdefence.org was only referenced 12 times in the Pre-Peak period, compared to 238 in the Peak period.

**Influential authors.** Table 6 shows the top 5 most influential Authors for the Residual and Low-Credibility datasets in the COVID and AZ Corpora using out-degree centrality as a measure of influence. The majority of the influential Authors predominantly post in English. Spanish language corporate media accounts feature as influential authors in both the Residual and Low-Credibility COVID datasets. For the AZ Corpus, English, French, German and Arabic language accounts feature as highly influential in the Residual dataset, but the influential Authors for low-credibility content are all English-speaking accounts.

Notably, as shown in Table 7, most of the influential Authors of low-credibility tweets are associated with low-credibility sources which feature in the Iffy+ List used to identify low-credibility domains for this research. Association is inferred by manually inspecting the user and comparing with those domains which feature in the Iffy+ list.

**Account properties.** The account properties for the most influential Authors from the Residual and Low-Credibility datasets, identified as the 100 Authors with the highest out-degree centrality, were compared to the account properties for the active Twitter userbase during the study period.

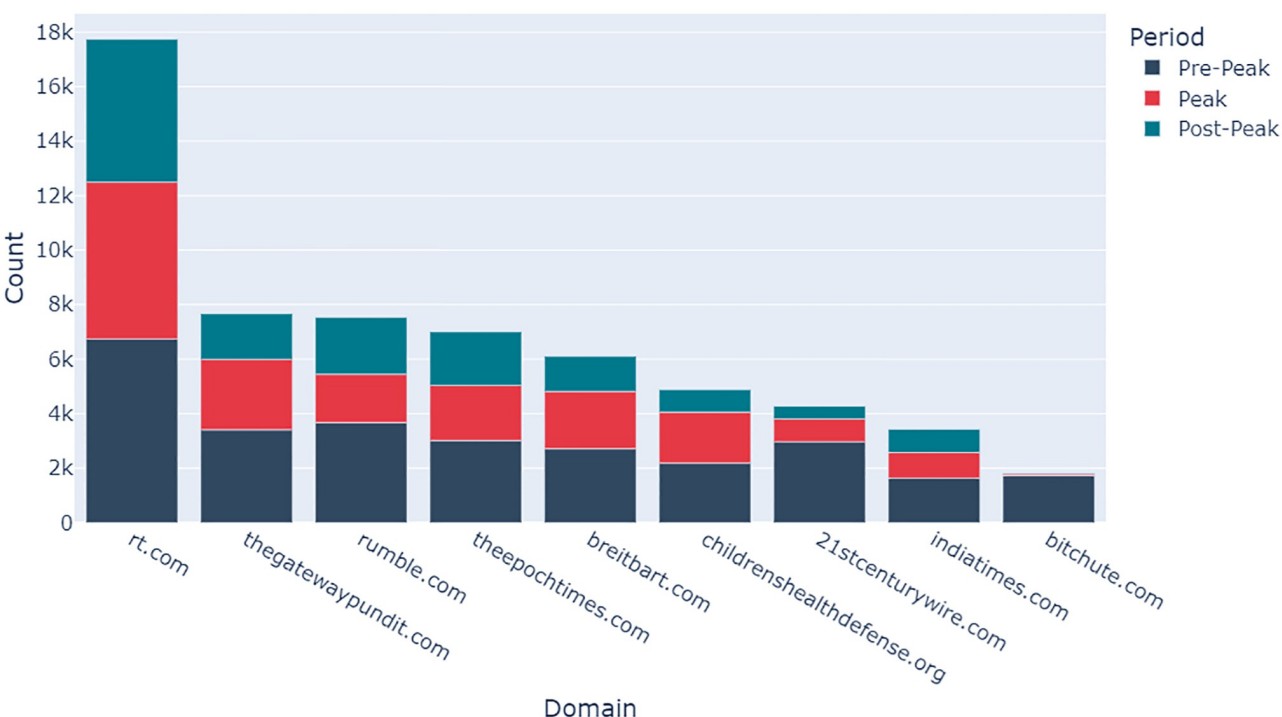

**Fig 10. Popular sources of low-credibility information from the COVID Corpus based on the cumulative number of times they were tweeted and retweeted during the Pre-Peak (navy), Peak (red), and Post-Peak (teal) periods.**

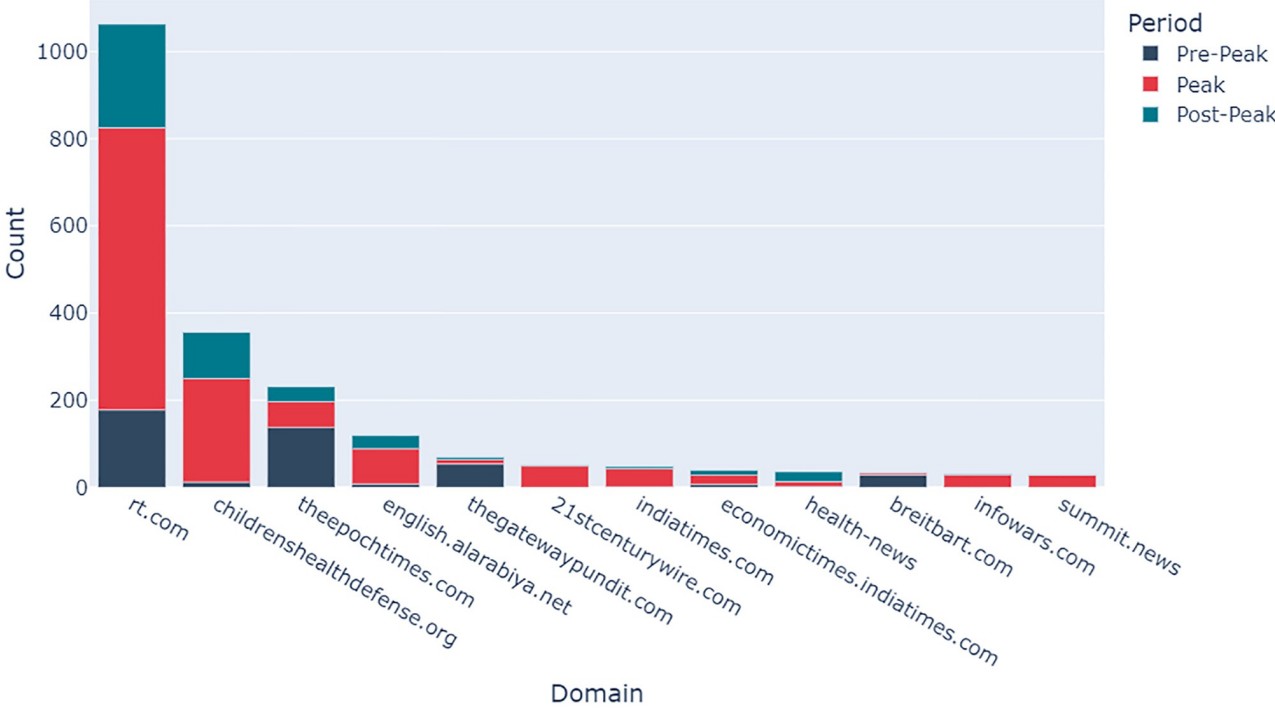

**Fig 11. Popular sources of low-credibility information from the AZ Corpus based on the cumulative number of times they were tweeted and retweeted during the Pre-Peak (navy), Peak (red), and Post-Peak (teal) periods.**

**Table 6. Most influential Authors in the Residual and low-credibility datasets using out-degree centrality as measures of influence.**

| | Author | Details |
|---|---|---|
| | **COVID Corpus, Residual dataset** | |
| 1 | KenRoth | Executive director of Human Rights Watch, an international NGO that conducts research and advocacy on human rights. Predominantly posts in English. |
| 2 | DrEricDing | Epidemiologist, health economist, and co-founder of the World Health Network, a coalition of citizens and experts set up in response to COVID-19. Predominantly posts in English. |
| 3 | SFMONEY_SFM | A peer-to-peer cryptocurrency. Predominantly posts in English. |
| 4 | CNN | A multinational news-based television channel headquartered in the US. Predominantly posts in English. |
| 5 | VTVcanal8 | A Venezuelan state-run television station. Predominantly posts in Spanish. |
| | **COVID Corpus, Low-Credibility dataset** | |
| 1 | 21WIRE | An account suspended for violating Twitter's COVID-19 misinformation policy. Co-founder and executive editor of 21st Century Wire, a conspiracy and conjecture site which features on the Iffy+ List. Predominantly posts in English. |
| 2 | DonaldJTrumpJr | The eldest child of Donald Trump, former president of the US. Predominantly posts in English. |
| 3 | EpochTimes | A far-right international multi-language newspaper and media company, which features on the Iffy+ List. Predominantly posts in English. |
| 4 | ActualidadRT | The Spanish language Twitter account for Russian government-funded television news network RT.com. RT.com features on the Iffy+ List. Predominantly posts in Spanish. |
| 5 | RobertKennedyJr | Founder and chairman of the Children's Health Defense, which features on the Iffy+ List. Predominantly posts in English. |
| | **AZ Corpus, Residual dataset** | |
| 1 | DrEricDing | *As above.* |
| 2 | Karl_Lauterbach | A scientist, physician, and politician of the Social Democratic Party of Germany who has served as Federal Minister for Health since December 2021. Predominantly posts in German. |
| 3 | momotchiii | A suspended account, which previously posted details of protests against COVID-19 restrictions in France and shared anti-vaccine content. Predominantly posts in French. |
| 4 | SandraWeeden | A suspended account, which previously posted a range of news articles about adverse reactions to COVID-19 vaccines. Predominantly posts in English. |
| 5 | OKAZ_online | A Saudi Arabian daily newspaper. Predominantly posts in Arabic. |
| | **AZ Corpus, Low-Credibility dataset** | |
| 1 | RT_com | A Russian government-funded television news network which features on the Iffy+ List. Predominantly posts in English. |
| 2 | RobertKennedyJr | *As above.* |
| 3 | PatriotOutfitrs | A suspended account, which previously posted news articles asking for President Biden to be impeached. Predominantly posts in English. |
| 4 | AlArabiya_Eng | The English language Twitter account for international Arabic television news channel Al Arabiya. Al Arabiya features on the Iffy+ List. Predominantly posts in English. |
| 5 | EconomicTimes | An English-language Indian daily newspaper the Economic Times which features on the Iffy+ List. Predominantly posts in English. |

The proportion of verified accounts amongst the Userbase, the Authors, and the influential Authors are shown in Table 8.

A greater proportion of Authors are verified than the Userbase. For both the Userbase and Authors, the Residual dataset contains the highest percentage of verified accounts, followed by the Low-Credibility dataset, followed by the Full dataset. There is a substantially higher proportion of the top 100 most influential Authors who are verified compared to all Authors, with notably more verified Influential Residual Authors than verified Influential Low-Credibility Authors.

**Table 7. Association between the most influential Authors in the low-credibility datasets and the domains which feature in the Iffy+ List.**

|  |  | Author | Associated with domain in Iffy+ List |
|---|---|---|---|
| COVID Corpus | 1 | 21WIRE | Yes, 21st Century Wire |
|  | 2 | DonaldJTrumpJr | No |
|  | 3 | EpochTimes | Yes, the Epoch Times |
|  | 4 | ActualidadRT | Yes, RT |
|  | 5 | RobertKennedyJr | Yes, the Children's Health Defense |
| AZ Corpus | 1 | RT_com | Yes, RT |
|  | 2 | RobertKennedyJr | Yes, the Children's Health Defense |
|  | 3 | PatriotOutfitrs | No |
|  | 4 | AlArabiya_Eng | Yes, Al Arabiya |
|  | 5 | EconomicTimes | Yes, the Economic Times |

**Table 8. Proportion of dataset who have verified accounts.**

|  | Full Dataset | Residual Dataset | Low-Credibility Dataset |
|---|---|---|---|
| Userbase | 0.91% | 3.0% | 1.5% |
| Authors | 1.8% | 5.5% | 2.6% |
| Top 100 influential Authors |  | 72% | 39% |

Fig 12 shows a comparison of the account creation dates for the Userbase with the influential Residual and Low-Credibility Authors. Pair-wise comparisons of each of the datasets using the Mann-Whitney U test indicate statistically significant differences between the account creation dates for each (p-value < 0.05). There is a notable peak in all three histograms in 2020, the year the pandemic started. 25% of the Userbase use accounts created since 11th March 2020, the date the WHO declared COVID-19 a pandemic. This compares to 18% for the Influential Low-Credibility Authors, and 11% for the Influential Residual Authors.

Fig 13 shows the distribution of the number of followers associated with each of the datasets. Again, a pairwise comparison using the Mann-Whitney U test indicates statistically significant differences between each dataset (p-value < 0.05). The Influential Residual Authors have a median follower count of nearly 550,000 compared to 58,000 for the Influential Low-Credibility Authors, and just 200 for the Userbase.

## Discussion

### Information diffusion changes

The research identified a correlation between the peak of media coverage linking the AZ vaccine to blood clots, and the proportion of AZ-related tweets which included URLs and low-credibility information. The research also identified a greater proportion of low-credibility AZ-related tweets were retweeted during and after the peak of media coverage than beforehand, with no significant changes in any other dataset observed. These findings indicate that not only are more people posting AZ-related content during peak of negative news coverage, but more of those tweets were proliferating low-credibility content, and there was increased engagement by Twitter uses with the low-credibility content.

From the data analysed within this research, it is not possible to draw conclusions about whether the increased volume of low-credibility AZ-related content being posted and shared on Twitter is due to an increased volume of low-credibility content online, a concerted effort

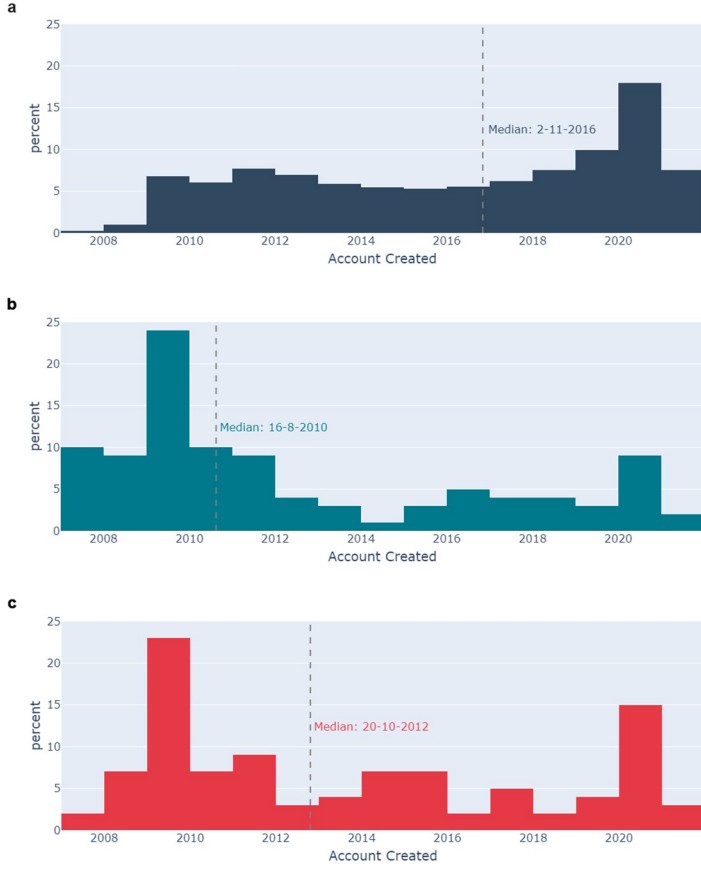

**Fig 12. Histograms showing the distribution account creation dates for 12a the Userbase (navy), 12b the Influential Residual Authors (teal), and 12c the Influential Low-Credibility Authors (red).**

by the producers of such content to promote it, or solely increased engagement by Twitter users with available content.

These observations indicate that the increased negative news coverage regarding the AZ vaccine had an impact on the number of AZ-related tweets, in particular with tweets containing URLs associated with low-credibility sources. However, the negative news coverage had no notable impact on the propagation of COVID-19-related information or misinformation on Twitter. There was no increase in the volume or potential exposure of COVID-19-related tweets, and the speed at which these tweets were shared remained consistent across the three time periods.

Findings from the centrality measures identified greater levels of interconnectivity prior to the peak media coverage for low-credibility AZ-related content. This was reflected in the smaller clusters of Sharers retweeting a larger number of Authors' content in the low-credibility networks during and after the AZ vaccine media coverage peaked. This is indicative of smaller, more disconnected discussions around a larger number of topics. There were no other notable changes to network properties or structures observed between the three time periods which could be correlated with the breaking AZ vaccine news in the media.

This research provides a valuable insight into the propagation of low-credibility information relating to the subject of a breaking news events. Whilst there was no net increase in the volume of COVID-19-related misinformation, the increase in AZ-related low-credibility

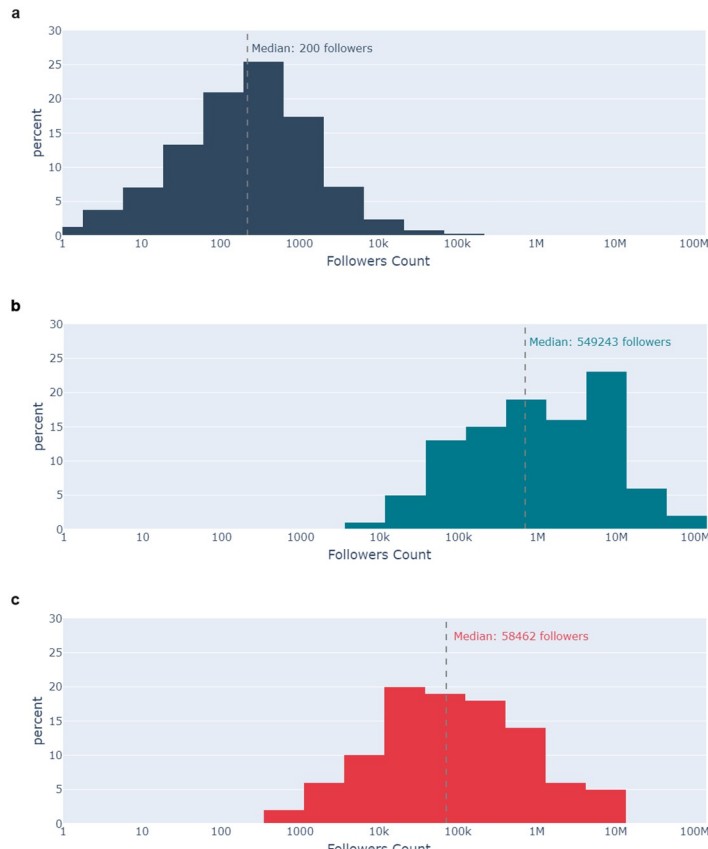

**Fig 13. Histograms showing the distribution of follower count for 13a the Userbase (navy), 13b the Influential Residual Authors (teal), and 13c the Influential Low-Credibility Authors (red).**

information in absolute and proportional terms indicates that disseminators of such content are guided by major news events. This finding will be valuable for public health agencies and owners of social media platforms alike, as it suggests that the same methods developed to tackle COVID-19-related misinformation generally will likely be applicable in the aftermath of breaking news. However, the research provides evidence that a targeted campaign to counter subject-specific misinformation would likely be effective in reducing the volume and exposure at times of heightened negative media coverage.

## Influential domains and users

This research has identified both the most influential domains and users in the propagation of low-credibility COVID-19-related misinformation on Twitter. Details of the most popular low-credibility domains provides a valuable insight into the source of COVID-19-related mis-information and a start-point for addressing it. RT.com is a dominant source of low-credibility content, constituting nearly 20% of all low-credibility COVID-19-related URLs, and over 40% of all low-credibility AZ-related URLs shared in the study period. Therefore, reducing the propagation of content from RT.com would have substantially reduced the overall volume of low-credibility content circulating on Twitter in March 2021. The simple methodology used to identify popular sources of low-credibility information could be deployed by social media

platforms and public health authorities in near-real-time to facilitate directed action to reduce the spread of misinformation on Twitter.

The most influential accounts posting about COVID-19 and AZ during the study period posted in a range of languages: Spanish, Arabic, French, German, and, most commonly, English. Portuguese and Chinese were the only two languages which were represented in the keyword list used to create the COVID Corpus, but were not featured in the list of influential accounts. English language accounts dominated the influential list for the COVID Corpus, indicating the dominance of English language COVID-19-related content on Twitter during this time. Notably, whilst the five most influential accounts posting about AZ did so in four different languages, all five of the most influential accounts posting low-credibility content about AZ were English-speaking. Further research would be required to understand why low-credibility content was more prevalent, in this instance, on English-speaking accounts.

Both corporate Twitter profiles associated with low-credibility media outlets, and personal Twitter profiles associated with the founders of such organisations, are influential in the propagation of low-credibility information (illustrated in Table 7). This close correlation between the most influential Twitter users and sources of low-credibility information indicates that a resource-efficient means to address misinformation once a low-credibility domain has been identified would be to explore the content being promoted via any associated Twitter accounts.

As the structure of the networks produced in this analysis are reminiscent of the broadcast networks identified by Smith et al. [23], prominent news and media organisations would be expected to feature as the most influential Authors in the Residual datasets. However, only three of the most influential Authors are corporate accounts for media outlets (@CNN, @VTVcanal8, and @OKAZ_online). Three of these profiles (@KenRoth, @DrEricDing, and @Karl_Lauterbach) are personal accounts associated with authorities on health- and human rights-related matters. These results suggest that individuals can be as influential in public discourse on Twitter as corporate media accounts.

The research identified significant differences between the profiles of Influential Authors and the Userbase. Influential Authors were much more likely to be verified (authenticated by Twitter as an account of public interest) than the Userbase. Notably, the Influential Low-Credibility Authors were less likely to be verified than the Influential Residual Authors. The findings regarding the number of followers reinforce expectations that influence is in-part correlated with audience size. The Influential Residual Authors had, on average, more than ten times as many followers as the Influential Low-Credibility Authors, and over 2500 more followers than the Userbase. These findings indicate a correlation between verified status, follower count, and influence, but that correlation is weaker in the context of low-credibility information.

Finally, the research identified that the Influential Authors were typically using accounts which had been created earlier than the Userbase. Across all categories there is a spike in account creation in 2020, the year the pandemic started. For Influential Authors, this suggests that new accounts may have been created to post COVID-19-related content. For the Userbase, where this peak is most prominent, this could indicate that people saw Twitter as a means to access up-to-date information on COVID-19, or identified it as a means to share information on the topic.

## Opportunities and challenges

Misinformation poses a significant challenge for public health, particularly during a pandemic. Social media platforms provide easy mechanisms by which information can be shared without

a requirement for fact-checking or corroboration. This can have a damaging impact on public health outcomes.

The very mechanisms which make misinformation commonplace on social media can be harnessed to tackle it. Public health authorities should use their own social media and online presence to combat the spread of misinformation on Twitter, prioritising content tackling subject-specific misinformation in the aftermath of a breaking news event. Analysis could also be conducted into the most popular sources of low-credibility content, with mitigation strategies developed specifically to reduce the propagation and impact of those sources to curtail the dissemination of misinformation.

Notably, none of the most influential users in the Residual datasets were associated with public health authorities. These findings are reminiscent of findings by Ahmed et al., who identified that in the early stages of the COVID-19 pandemic there was a lack of an authority figure actively combating COVID-19 misinformation [25]. Additional research indicates that whilst scientific and health-related clusters on Twitter received significant attention at the start of the pandemic, by mid-2020 attention shifted towards national elites and political actors [26].

## Limitations and future research

### Limitations

As this research relies on the analysis of Twitter data, the results have potential to be generalised more widely. The demographic of Twitter users is not representative of the global population, with overrepresentation of younger people and men [27].

The use of a broad keyword list to identify COVID-19-related content may result in tweets being erroneously included within the dataset. This is highlighted in Fig 5 where a large volume of tweets relating to Nigerian actress Erica Ngozi Nlewedim were identified which do not relate to COVID-19.

The use of low-credibility online sources to determine whether a tweet was misinformation also has limitations. Whilst this methodology is widely adopted across the literature, it is solely able to categorise those tweets which contain URLs associated with domains which have been independently categorised as low-credibility. This approach almost certainly underestimates the scale of misinformation by omitting false or misleading information spread through the body of tweets or comments, or references to low-credibility sources which were not included on the Iffy+ List.

Twitter's information sharing model presented several limitations in the network analysis conducted in this research. As it does not preserve the information sharing chain, the role of intermediaries within information sharing pathways is neglected. This has restricted the types of network analysis which can be conducted and impacted on the interpretations of findings regarding network structure. Intermediaries play a pivotal role in the dissemination of information, and further work to reconstruct the true pathways would enable a more thorough investigation into any changes in information sharing structures correlated with major news events. This could reveal key nodes within the network which could be targeted by public health authorities or social media platform owners to curtail the spread of low-credibility content.

This research was able to capture the volume and potential exposure of COVID-19- and AZ-related information and misinformation but was not able to determine the actual exposure or measure the impact of that exposure. The methodology used follower lists as a proxy for the potential audience for a tweet. This is neither an upper nor lower bound of true exposure, as exposure could happen outside a follower relationship, and there is no guarantee that all

followers will see a particular tweet. Further, the methodology relied on retweets as an indicator of those users who were sufficiently influenced by the content to proactively share it with their followers. However, this does not capture those who were exposed and influenced by the content but chose not to retweet it.

The dataset used in this research was derived from a keywords list comprised of seven languages. However, many more countries had an AZ vaccination programme in March 2021, and tweets in those languages were not captured.

Finally, this research did not look at where the Twitter users were based as there is no robust location data in the dataset. Whilst COVID-19 was a global pandemic, the AZ vaccine was not used globally. Therefore, the links to blood clots were only significant in countries with an AZ vaccination programme. Greater insights could be generated by looking at the geographic spread of users within the COVID and AZ Corpora, or by filtering each corpus to only contain tweets originating from countries with an AZ vaccination programme.

## Future research

Based on the above limitations and discussion, there is significant opportunity for future research. Particularly, to further advance public heath authorities' ability to combat low-credibility information, research into the changes in the production and promotion of low-credibility content following a breaking news event would be valuable. Additionally, further work to rebuild the true information sharing pathways would provide an interesting insight into the role of intermediaries and the full extent of changes in network structures following a major news event. Finally, repeating this study in the context of other breaking news events will help determine the replicability of the findings.

## Conclusion

This study's findings indicate that there was no significant change to the volume or propagation of COVID-19-related information or misinformation on Twitter in response to the breaking news linking the AZ vaccine to blood clots. However, the research did identify several changes in the volume and propagation of AZ-related information, and particularly low-credibility information, which correlates with the peak news coverage. This suggests that the methods developed to tackle COVID-19-related misinformation will likely remain effective in the aftermath of breaking news. However, a more targeted approach to AZ-related misinformation would be a valuable addition to the toolkit for countering low-credibility information in the immediate aftermath of the breaking news.

The research identified RT.com as a prolific source of low-credibility COVID-19- and AZ-related content on Twitter. The high proportion of low-credibility content shared which derived from RT.com suggests that a targeted campaign to alert Twitter users to the questionable quality of RT.com content, and to counter misinformation authored by the network could have a substantial impact on reducing the flow of COVID-19-related misinformation on Twitter. The methodology used to identify the most prominent low-credibility sources could easily be applied in near-real-time by authorities or social media platforms, providing them with valuable insights into where to target specific mitigation activities.

The research further identified the significant role of low-credibility media outlets self-promoting their content in the successful propagation of low-credibility content. This indicates that a resource-efficient means to tackle misinformation on Twitter would be to identify the most popular low-credibility domains and consider both the corporate Twitter profiles associated with media outlet and any personal Twitter profiles associated with the founders as likely influential authors of low-credibility content.

The study identified a notable absence of authority figures actively combating COVID-19 misinformation on Twitter. As outlined above, the totality of the findings provides insights which can helpfully inform actionable strategies for minimising the spread and reach of COVID-19-related misinformation. Adoption of these approaches by public health authorities could help reduce the volume and impact of COVID-19-related misinformation generally, and in the immediate aftermath of breaking news events.

## Supporting information

**S1 Table. Keywords used in the Twitter API to create the COVID Corpus.**
(DOCX)

**S1 Appendix. Keywords used to create the AZ Corpus.**
(DOCX)

**S1 File. Residual COVID dataset, Pre-Peak period, file 1 of 16.**
(CSV)

**S2 File. Residual COVID dataset, Pre-Peak period, file 2 of 16.**
(CSV)

**S3 File. Residual COVID dataset, Pre-Peak period, file 3 of 16.**
(CSV)

**S4 File. Residual COVID dataset, Pre-Peak period, file 4 of 16.**
(CSV)

**S5 File. Residual COVID dataset, Pre-Peak period, file 5 of 16.**
(CSV)

**S6 File. Residual COVID dataset, Pre-Peak period, file 6 of 16.**
(CSV)

**S7 File. Residual COVID dataset, Pre-Peak period, file 7 of 16.**
(CSV)

**S8 File. Residual COVID dataset, Pre-Peak period, file 8 of 16.**
(CSV)

**S9 File. Residual COVID dataset, Pre-Peak period, file 9 of 16.**
(CSV)

**S10 File. Residual COVID dataset, Pre-Peak period, file 10 of 16.**
(CSV)

**S11 File. Residual COVID dataset, Pre-Peak period, file 11 of 16.**
(CSV)

**S12 File. Residual COVID dataset, Pre-Peak period, file 12 of 16.**
(CSV)

**S13 File. Residual COVID dataset, Pre-Peak period, file 13 of 16.**
(CSV)

**S14 File. Residual COVID dataset, Pre-Peak period, file 14 of 16.**
(CSV)

**S15 File. Residual COVID dataset, Pre-Peak period, file 15 of 16.**
(CSV)

**S16 File. Residual COVID dataset, Pre-Peak period, file 16 of 16.**
(CSV)

**S17 File. Residual COVID dataset, Peak period, file 1 of 15.**
(CSV)

**S18 File. Residual COVID dataset, Peak period, file 2 of 15.**
(CSV)

**S19 File. Residual COVID dataset, Peak period, file 3 of 15.**
(CSV)

**S20 File. Residual COVID dataset, Peak period, file 4 of 15.**
(CSV)

**S21 File. Residual COVID dataset, Peak period, file 5 of 15.**
(CSV)

**S22 File. Residual COVID dataset, Peak period, file 6 of 15.**
(CSV)

**S23 File. Residual COVID dataset, Peak period, file 7 of 15.**
(CSV)

**S24 File. Residual COVID dataset, Peak period, file 8 of 15.**
(CSV)

**S25 File. Residual COVID dataset, Peak period, file 9 of 15.**
(CSV)

**S26 File. Residual COVID dataset, Peak period, file 10 of 15.**
(CSV)

**S27 File. Residual COVID dataset, Peak period, file 11 of 15.**
(CSV)

**S28 File. Residual COVID dataset, Peak period, file 12 of 15.**
(CSV)

**S29 File. Residual COVID dataset, Peak period, file 13 of 15.**
(CSV)

**S30 File. Residual COVID dataset, Peak period, file 14 of 15.**
(CSV)

**S31 File. Residual COVID dataset, Peak period, file 15 of 15.**
(CSV)

**S32 File. Residual COVID dataset, Post-Peak period, file 1 of 14.**
(CSV)

**S33 File. Residual COVID dataset, Post-Peak period, file 2 of 14.**
(CSV)

**S34 File. Residual COVID dataset, Post-Peak period, file 3 of 14.**
(CSV)

**S35 File. Residual COVID dataset, Post-Peak period, file 4 of 14.**
(CSV)

**S36 File. Residual COVID dataset, Post-Peak period, file 5 of 14.**
(CSV)

**S37 File. Residual COVID dataset, Post-Peak period, file 6 of 14.**
(CSV)

**S38 File. Residual COVID dataset, Post-Peak period, file 7 of 14.**
(CSV)

**S39 File. Residual COVID dataset, Post-Peak period, file 8 of 14.**
(CSV)

**S40 File. Residual COVID dataset, Post-Peak period, file 9 of 14.**
(CSV)

**S41 File. Residual COVID dataset, Post-Peak period, file 10 of 14.**
(CSV)

**S42 File. Residual COVID dataset, Post-Peak period, file 11 of 14.**
(CSV)

**S43 File. Residual COVID dataset, Post-Peak period, file 12 of 14.**
(CSV)

**S44 File. Residual COVID dataset, Post-Peak period, file 13 of 14.**
(CSV)

**S45 File. Residual COVID dataset, Post-Peak period, file 14 of 14.**
(CSV)

**S46 File. Low-Credibility COVID dataset, Pre-Peak period.**
(CSV)

**S47 File. Low-Credibility COVID dataset, Peak period.**
(CSV)

**S48 File. Low-Credibility COVID dataset, Post-Peak period.**
(CSV)

**S49 File. Residual AZ dataset, Pre-Peak period.**
(CSV)

**S50 File. Residual AZ dataset, Peak period.**
(CSV)

**S51 File. Residual AZ dataset, Post-Peak period.**
(CSV)

**S52 File. Low-Credibility AZ dataset, Pre-Peak period.**
(CSV)

**S53 File. Low-Credibility AZ dataset, Peak period.**
(CSV)

**S54 File. Low-Credibility AZ dataset, Post-Peak period.**
(CSV)

## Author Contributions

**Conceptualization:** Ali Hobbs.

**Data curation:** Ali Hobbs, Aisha Aldosery, Patty Kostkova.

**Investigation:** Ali Hobbs.

**Methodology:** Ali Hobbs.

**Supervision:** Aisha Aldosery, Patty Kostkova.

**Writing – review & editing:** Aisha Aldosery, Patty Kostkova.

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
