## [Decision Letter · Decision Letter 0]

31 Aug 2023

PONE-D-23-16069Identifying how the diffusion of low-credibility COVID-19-related information on Twitter reacts to reports linking rare blood clots with the Oxford/AstraZeneca COVID-19 vaccinePLOS ONE

Dear Dr. Hobbs,

Thank you for submitting your manuscript to PLOS ONE. After careful consideration, we feel that it has merit but does not fully meet PLOS ONE’s publication criteria as it currently stands. Therefore, we invite you to submit a revised version of the manuscript that addresses the points raised during the review process.

We look forward to receiving your revised manuscript.

Kind regards,

Pierluigi Vellucci

Academic Editor

PLOS ONE

Journal Requirements:

2. In your Methods section, please include additional information about your dataset and ensure that you have included a statement specifying whether the collection and analysis method complied with the terms and conditions for the source of the data.

3. Please upload a new copy of Figures 5a and 5b  as the detail is not clear. Please follow the link for more information: " ext-link-type="uri" xlink:type="simple">https://blogs.plos.org/plos/2019/06/looking-good-tips-for-creating-your-plos-figures-graphics/"
https://blogs.plos.org/plos/2019/06/looking-good-tips-for-creating-your-plos-figures-graphics

Reviewers' comments:

Reviewer's Responses to Questions

**Comments to the Author**

1. Is the manuscript technically sound, and do the data support the conclusions?

Reviewer #1: Partly

Reviewer #2: Yes

2. Has the statistical analysis been performed appropriately and rigorously? 

Reviewer #1: Yes

Reviewer #2: Yes

3. Have the authors made all data underlying the findings in their manuscript fully available?

Reviewer #1: Yes

Reviewer #2: Yes

4. Is the manuscript presented in an intelligible fashion and written in standard English?

Reviewer #1: Yes

Reviewer #2: Yes

5. Review Comments to the Author

Reviewer #1: Title: I think that a shorter title could help to understand what the study is about.

Line 2: Readers in the future may be not so familiar with the term COVID-19 alone. Authors could introduce the fact that there has been a pandemic, or at least write “The COVID-19 pandemic has occurred etc”.

Line 50: I think it is important to check and report whether these websites verify sources in all the different languages that you are collecting. Otherwise, some low-credibility sources may be neglected only because the Iffy dataset did not do any verification at all for websites in some specific languages. In this, case this should be properly enunciated in the limitations of the study.

Line 83 (and elsewhere): "network" and "graph" synonyms, thus the usage of "network graph" is improper. I would recommend using only the term "network", consistently along the paper.

Line 89: Authors should specify here which is the direction of the edge

Line 201 and Table 2: When an average quantity is shown, also standard deviation could be shown alongside (e.g. average +- std), to help understand when a quantity is greater or lower than another in the absence of statistical tests.

Line 214: since the authors are measuring the lag until the first retweet only, I am not sure that “speed of diffusion” is the most appropriate term. With “Speed” one usually refers to a quantity per time unit ( e.g. average number of retweets per day, or average potential exposure per day). I would consider other possible names if the authors have other ideas.

Line 253: Authors should mention in the Methods which algorithm was used for network layout (computations of coordinates). The interpretation of proximity in the plot depends on the algorithm chosen.

Line 289: the density of a network can be easily computed. Authors could report a numeric result instead that relying on a figure.

Figure 14-19: for some reason, the pdf file I have been provided does not contain images from 14 onwards. For this reason, I cannot review them for now.

Line 360: Authors could specify that here "association" is inferred by manually inspecting the user, and it is not an association with preferential sharing of a particular source.

Line 407-409: If I understood well, the correlation is not only with low-credibility, but with AZ-related tweets in general. I would rephrase to say that the correlation is with AZ-related tweets in general, and in particular with tweets containing URLs associated with low-credibility sources.

Line 425-428: I am not sure that the authors are providing numerical results supporting this statement (e.g. community detection/clustering), and I don't think the figures alone can support your statement. Therefore, I suggest removing these two sentences from the final discussion, or at least moving them as an additional comment (not a proper result), after the next sentence where you refer to the degree centrality results. Alternatively, authors can perform additional analyses to quantify the variation of clusters.

Line 466: typo: “would be expected TO feature”

Line 470-471: This sentence is not very clear in English, at least to me. I suggest to rephrase it.

Line 502: missing reference to Ahmed et al.

Line 504: Interesting comment. It has also been shown that in the very first months of the pandemic, the international scientific community lost its audience on Twitter, and national political communities became the most prominent authors (Clusters of science and health related Twitter users become more isolated during the COVID-19 pandemic, Sci Rep 2021). Authors dispose of a rich multilingual dataset but do not explicitly use multilingualism to add some richness to the results. I think the authors may add some insights to the mentioned study about the "nationalization" of the debate, by showing the prominent language of each author in Table 6 (e.g. KenRoth (eng) ) and comment appropriately.

Reviewer #2: The manuscript is well-written, well-presented, and the research conducted within it is interesting and, to my knowledge, new. For these reasons, I recommend it for publication, even if there are some points that I would like to be clarified or improved. Please see the attached file for the aforementioned questions.

6. PLOS authors have the option to publish the peer review history of their article (what does this mean?). If published, this will include your full peer review and any attached files.

Reviewer #1: No

Reviewer #2: No

---

## [Author Response · Author response to Decision Letter 0]

19 Oct 2023

Responses to all commenst have been included in the "Response to Reviewers" file.

---

## [Decision Letter · Decision Letter 1]

13 Dec 2023

Low credibility URL sharing on Twitter during reporting linking rare blood clots with the Oxford/AstraZeneca COVID-19 Vaccine

PONE-D-23-16069R1

Dear Dr. Hobbs,

We’re pleased to inform you that your manuscript has been judged scientifically suitable for publication and will be formally accepted for publication once it meets all outstanding technical requirements.

Kind regards,

Pierluigi Vellucci

Academic Editor

PLOS ONE

Additional Editor Comments (optional):

Reviewers' comments:

Reviewer's Responses to Questions

**Comments to the Author**

1. If the authors have adequately addressed your comments raised in a previous round of review and you feel that this manuscript is now acceptable for publication, you may indicate that here to bypass the “Comments to the Author” section, enter your conflict of interest statement in the “Confidential to Editor” section, and submit your "Accept" recommendation.

Reviewer #1: All comments have been addressed

Reviewer #2: All comments have been addressed

2. Is the manuscript technically sound, and do the data support the conclusions?

Reviewer #1: Yes

Reviewer #2: Yes

3. Has the statistical analysis been performed appropriately and rigorously? 

Reviewer #1: Yes

Reviewer #2: Yes

4. Have the authors made all data underlying the findings in their manuscript fully available?

Reviewer #1: Yes

Reviewer #2: Yes

5. Is the manuscript presented in an intelligible fashion and written in standard English?

Reviewer #1: Yes

Reviewer #2: Yes

6. Review Comments to the Author

Reviewer #1: (No Response)

Reviewer #2: Please double-check for typos the reference: you wrote "Agnosti F, Toffolutti V, Cavalli N, Nivakoski S, Mascherini M, Aassve A.

Information and vaccine hesitancy: Evidence from the early stage of the vaccine

roll-out in 28 European countries. Plos one. 2022; 17(9)."

7. PLOS authors have the option to publish the peer review history of their article (what does this mean?). If published, this will include your full peer review and any attached files.

Reviewer #1: No

Reviewer #2: No

---

## [Editor Report · Acceptance letter]

8 Jan 2024

PONE-D-23-16069R1 

PLOS ONE

Dear Dr. Hobbs, 

I'm pleased to inform you that your manuscript has been deemed suitable for publication in PLOS ONE. Congratulations! Your manuscript is now being handed over to our production team.

Kind regards, 

on behalf of

Dr. Pierluigi Vellucci 

Academic Editor

PLOS ONE